# COLLABORATIVE CONCEPT DRIFT DETECTION

**Beverly Abadines Quon & Jean-Luc Gaudiot**
Electrical Engineering and Computer Science Department
University of California, Irvine
Irvine, CA 92697
{babadine, gaudiot}@uci.edu

## ABSTRACT

Collaborative Concept Drift Detection (C2D2) combines Fast Correlated Based Filtering (FCBF) and Singular Value Decomposition (SVD) to detect concept drifts in 5 synthetic datasets. We compare our results against 6 diveregence tests and introduce Performance Gain Update Cost Ratio (PGUCR). Post-hoc Tukey HSD test confirmed that C2D2 outperformed the other tests in terms of PGUCR. Much of C2D2's improvement is based on its conservative signals for updates.

## 1 CONCEPT DRIFT

Machine learning models train on data with the expectation that the **concept** or relationship discerned between the predictors and target variable are consistent with post training data (e.g. online data streams). Data streams, however, are unstationary and can result in **concept drifts**, where said relationship no longer holds (Widmer & Kubat (1996); Huyen (2022)). Consequently, concept drifts can result in model performance degradation despite the fact that the model itself is unchanged (Lu et al. (2019)). Additionally, models incur an update cost when they are retrained upon detection of drift. Although there are many types of drift detectors, data distribution or divergence tests can provide more explainability than performance based detectors. (Lu et al. (2018)). This paper extends a method originally used for feature selection in order to detect concept drift. We compare our findings to divergence tests and formulate a metric that relates the F1 obtained with the cost of retraining.

## 2 COLLABORATIVE CONCEPT DRIFT DETECTION

**Collaborative Concept Drift Detection (C2D2)** applies a window based **Fast Correlated Based Filtering (FCBF)** ( Yu & Liu (2003); Nguyen et al. (2012)), a multivariate feature selection method that considers both the class relevance and the dependency between each feature pair through the **Symmetrical Uncertainty (SU)**[3] computed from entropy.

Rather than remove the redundant features directly, the calculated SU are fed as a matrix, $A$ for Singular Value Decomposition (SVD) [1] where $V^T$ relates to batches and $U$ corresponds to the features. Taking only the top four $V^T$ and $U$ components noted by $S$, we calculate sum of the stepwise difference of $V^T$ for every batch. The resulting argmax and values within 1 standard deviation of the max are signalled as batches with drift.

$$A = USV^T \tag{1}$$

## 3 EXPERIMENT

C2D2 was tested on 4 artificial data sets generated by the MOA framework (Massive On-line Analysis) (Bifet et al. (2010; 2013)). Concept drift was modeled on MOA by joining data streams as a weighted combination of distributions whose probability of an instance stemming from the new concept is defined by a sigmoid function. Each dataset contained 10K instances and were injected with concept drifts of widths from 0.5K to 4K instances. The midpoints of drift ranged from 1.5k to 7.5K. 10 tests were generated from each dataset by modifying the instantiation of streams via a random seed.

C2D2 was evaluated against 6 divergence tests: Cramer Von Mises (CMV), Energy Distance test (ED), Kolmogorov-Smirnov test (KS), Mann-Whitney U-test (MW), T test, and Wasserstein Distance (WD). The 6 tests were implemented in a fixed sliding window fashion, which split the data into 10 batches and compared the $i^{th}$ batch with the $i^{th-1}$ batch as its respective current and reference windows. Batches that were deemed significantly different were signalled for retraining. Hoeffding tree (HFT) was used as the base classifier and was prompted to retrain on the most recent batch according to the signals generated by the detectors. In addition to these test was an ablation test, where the HFT made predictions without any retraining.

### 3.1 PERFORMANCE GAIN TO UPDATE COST RATIO

An overly cautious and ineffective method would arbitrarily signal for an update at every batch. To take in consideration the computational and temporal debt of retraining with respect to the performance gained, we introduce the **Performance Gain to Update Cost Ratio (PGUCR)** 2. Ratios of 0 are ineffective detectors and 1 are effective detectors. $F1_{new}$ is the $F1$ score of the base classifier that was retrained according to the signals, while $F1_{ablation}$ is the score without any retraining. $N_{update}$ is the number of batches signalled (i.e. max is 9 as the first batch is used for training). $Cost_{update}$ which can be seen as a penalty for updating is an adjustable parameter whose value is related to the importance of improving the $F1$ score. Our experiments set $Cost_{update}$ to 0.1.

$$PGUCR = \frac{1}{2}\left(1 + \frac{F1_{\text{new}} - F1_{\text{ablation}}}{F1_{\text{ablation}}}\right) / \left(1 + N_{\text{update}} \times Cost_{\text{update}}\right) \qquad (2)$$

## 4 RESULTS AND CONCLUSION

ANOVA indicated that there is a difference in mean values of PGUCR amongst the tests. Post-hoc Tukey HSD test confirmed that C2D2 provided significant improvement. Much of C2D2's improvement is based on the fact that it signalled for updates conservatively in comparison to the other tests[2], thereby decreasing its false positive rate. In contrast to the other tests, which monitored whether a feature's distribution had changed in comparison to itself, C2D2 was able to hint at the relationship of how the features have drifted collectively [A]. Future work should identify whether the collective nature of C2D2 can be applied to bringing explainability towards formerly independent models with overlapping feature spaces.

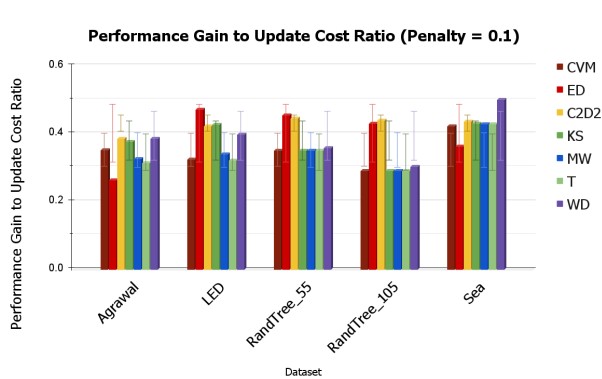

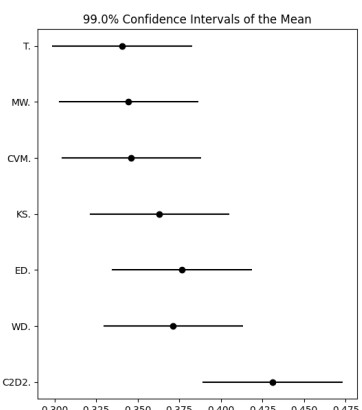

Figure 1: Mean PGUCR values with penalty of 0.1. Each dataset contained the following number of features: Agrawal (9), LED (24), RandTree_55 (55), RandTree_105 (105), Sea (4).

Figure 2: Post-hoc Tukey HSD test at 99% confidence interval indicates that C2D2 is significantly better than T, MW, CVM, KS, ED, and WD. C2D2 has a mean of 0.43.

ACKNOWLEDGEMENTS

This work was supported by Graduate Assistance in Areas of National Need (GAANN) under award P200A10052 and by the National Science Foundation under award CCF-176379.

URM STATEMENT

Author Quon meets the URM criteria of ICLR 2023 Tiny Papers Track.

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

## A  APPENDIX

**Symmetrical Uncertainty**: Calculates the mutual dependencies of random variables $X$ and $Y$ such that values closer to 0 indicate independence while values closer to 1 indicate dependence, where knowledge of one can predict the outcome of its pair. $IG$ is the information gain of $X$ given $Y$ and $H(X), H(Y)$ are entropies of $X, Y$.

$$SU(X,Y) = \frac{2 \times IG(X|Y)}{H(X) + H(Y)} \tag{3}$$

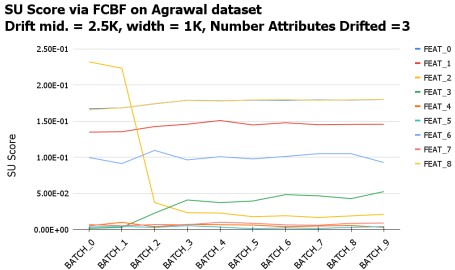

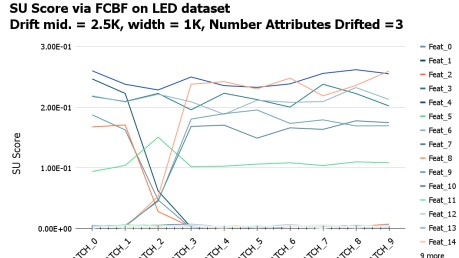

Figure 3: SU scores per feature per batch on one of the Agrawal dataset. Drift occurs from instances 2K to 3K (i.e. batch 2). Feat 2 decreases while Feat 3 increases at a crossover point between batch 2 and 3.

Figure 4: SU scores per feature per batch on one of the LED dataset. Drift occurs from instances 2K to 3K (i.e. batch 2). Feat 0, 1, and 2 decrease while Feat 7, 8 and 9 increase at a crossover point at batch 2.

Table 1: Average number of signals for updates on synthetic datasets. Ablation excluded as the count will always be zero. Counts can range from 0 to 9.

| TEST | AGRAWAL | LED | RANDTREE_55 | RANDTREE_105 | SEA |
|------|---------|-----|-------------|--------------|-----|
| CVM | $4.3 \pm 0.9$ | $8.1 \pm 0.9$ | $9.0 \pm 0.9$ | $9.0 \pm 0.0$ | $2.2 \pm 0.8$ |
| ED | $9.0 \pm 0.0$ | $\mathbf{2.0 \pm 0.7}$ | $2.4 \pm 0.5$ | $2.0 \pm 1.1$ | $4.3 \pm 1.1$ |
| C2D2 | $3.1 \pm 1.4$ | $3.6 \pm 1.3$ | $\mathbf{2.6 \pm 1.1}$ | $\mathbf{1.8 \pm 0.6}$ | $1.7 \pm 0.5$ |
| KS | $3.5 \pm 1.8$ | $3.6 \pm 1.7$ | $9.0 \pm 0.0$ | $9.0 \pm 0.0$ | $2.0 \pm 1.0$ |
| MW | $5.4 \pm 0.5$ | $7.7 \pm 2.1$ | $9.0 \pm 0.0$ | $9.0 \pm 0.0$ | $2.2 \pm 1.0$ |
| T | $6.0 \pm 0.7$ | $8.3 \pm 0.7$ | $9.0 \pm 0.0$ | $9.0 \pm 0.0$ | $2.2 \pm 1.0$ |
| WD | $\mathbf{3.0 \pm 0.7}$ | $4.7 \pm 0.9$ | $8.5 \pm 0.7$ | $8.0 \pm 0.8$ | $\mathbf{0.2 \pm 0.4}$ |

Table 2: Average F1 score on the datasets with 10 samples. The F1 scores of the tests, ablation (ABL), and number of signals are used to calculate PGUCR.

| TEST | AGRAWAL | LED | RANDTREE_55 | RANDTREE_105 | SEA |
|------|---------|-----|-------------|--------------|-----|
| CVM | $1.00 \pm 0.00$ | $0.66 \pm 0.04$ | $0.76 \pm 0.01$ | $0.67 \pm 0.03$ | $0.84 \pm 0.02$ |
| ED | $1.00 \pm 0.00$ | $0.64 \pm 0.05$ | $0.65 \pm 0.03$ | $0.63 \pm 0.05$ | $0.84 \pm 0.01$ |
| C2D2 | $1.00 \pm 0.00$ | $0.65 \pm 0.05$ | $0.64 \pm 0.04$ | $0.63 \pm 0.03$ | $0.84 \pm 0.01$ |
| KS | $1.00 \pm 0.00$ | $0.65 \pm 0.04$ | $0.76 \pm 0.01$ | $0.67 \pm 0.03$ | $0.84 \pm 0.02$ |
| MW | $1.00 \pm 0.00$ | $0.66 \pm 0.04$ | $0.76 \pm 0.01$ | $0.67 \pm 0.03$ | $0.83 \pm 0.02$ |
| T | $1.00 \pm 0.00$ | $0.66 \pm 0.04$ | $0.76 \pm 0.01$ | $0.67 \pm 0.03$ | $0.83 \pm 0.02$ |
| WD | $1.00 \pm 0.00$ | $0.66 \pm 0.04$ | $0.76 \pm 0.01$ | $0.66 \pm 0.04$ | $0.83 \pm 0.02$ |
| ABL | $1.00 \pm 0.00$ | $0.58 \pm 0.10$ | $0.58 \pm 0.07$ | $0.61 \pm 0.05$ | $0.83 \pm 0.02$ |

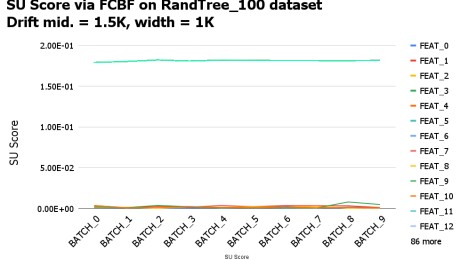

Figure 5: SU scores per feature per batch on one of the RandTree_55 dataset. Drift occurs from instances 1.5 K to 2.5K (i.e. batches 1 and 2)

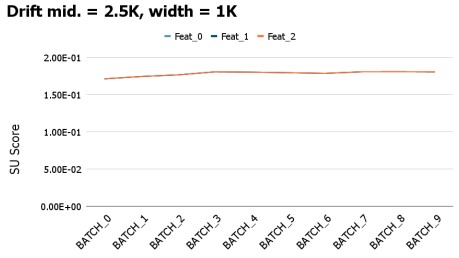

Figure 6: SU scores per feature per batch on one of the RandTree_100 dataset. Drift injected from instances 1 K to 2K (i.e. batch 1

Figure 7: SU scores per feature per batch on one of the Sea dataset. Drift injected from instances 2 K to 3K (i.e. batches 2)

