# OpenReview forum: "COLLABORATIVE CONCEPT DRIFT DETECTION"
_ICLR.cc/2023/TinyPapers — Submitted to Tiny Papers @ ICLR 2023_

### Official Review · Reviewer_LAyt · 2023-03-20

**Confidence:** 3

**Summary Of Contributions:**

The authors propose C2D2 (Collaborative Concept Drift Detection) to detect invariant patterns that could relate to drift. First, they map features and target spaces into a correlation matrix through Fast Correlation Based Filtering. Then, SVD is used to get the invariant representation and finally a post-hoc is used to indicate the occurrence of drift.

**Rating:**

Great Start (GS): a submission which meets some of the reviewing criteria but has room for improvement

**Strengths And Weaknesses:**

## Weaknesses:
1. Several phrases in the paper are difficult to follow or not entirely correct:
    1.  Section 1: *"Although the models themselves operate independently of each other, and they hold information that describes the relationship..."*
    2. Section 2: *"Similarly, our notion is to learn from multiple models in order to capture evolving concepts, but rather than triggering drift detection based on similarities between models-to-features and model-to-model relationships ..."*
    3. Section 2.1: *"To fit a model is to take a function f onto a feature vector, x such that it provides ...".* The comma is not needed here. Similarly Section 3.1: *"The correlation of every combination of x within X mapped to every y in Y is calculated by Fast Correlation Based Filtering (FCBF) ( Nguyen et al. (2012)) to form a correlation matrix, A..."*
2. The authors provide a comparison between C2D2 and the author's previous work which is anonymized and not provided in the paper. As a consequence, I cannot compare the two approaches.
3. Reproducibility: there are a few details that are missing in the paper to reproduce the results. What model and data are the authors using?

**Suggested Changes:**

As said in the "Weaknesses" section, I cannot compare C2D2 with the author's previous work.
More specifically, in Section 4 - Conclusion:
-  *"A limitation of XYZ is that its similarity scores are calculated via shapley values and the complexity explodes in complexity to O(2k ), where k is the entire feature size across all models. Additionally, it has an inherent trade-off between incurring high false alarms and long delays in detection."*
- *"With C2D2, however, the complexity is O(mn) and its constraint lies in its ability to identify how diverse
an ensemble needs to be in order to isolate the invariants to be related only to drift."* Why the complexity is  O(mn)?

---

### Official Review · Reviewer_d6SQ · 2023-03-28

**Confidence:** 3

**Summary Of Contributions:**

The author proposes C2D2 that relies on the diversity of models to identify invariant patterns related to drift. C2D2's complexity is lower than author’s previous model, and it focuses on capturing confounding variables related to drift by leveraging the ensemble feature space.

**Rating:**

High Potential (HP): a submission which meets the reviewing criteria and has potential to make an impact on the field

**Strengths And Weaknesses:**

## Strengths:

1. The proposed Collaborative Concept Drift Detection (C2D2) framework provides a new approach for detecting concept drift in ensemble models by leveraging the diversity of models and their feature spaces.
2. The research provides insights into the importance of considering overlapping feature spaces and predictors for drift detection in systems with multiple models.

## Weakness:

1. It is assumed the proposed C2D2 framework is evaluated on synthetic datasets, and its effectiveness on real-world datasets is unknown.
2.  The proposed method assumes that the confounding variables related to drift are invariant across the ensemble of classifiers, which may not always be the case.
3. The study does not provide a comprehensive comparison of the time complexity of C2D2 with other state-of-the-art concept drift detection methods.

**Suggested Changes:**

As a next step, the paper could benefit from further comparison with other state-of-the-art methods in terms of time complexity. This would provide a more comprehensive evaluation of the proposed method and help researchers to better understand its strengths and limitations in comparison to other methods.

Please consider expanding FCBF to Fast Correlation Based Filtering in the abstract.

---

### Author Response · Authors · 2023-05-31
**Revision update**

In response to promoting the correctness, clarity, and reproducibility of our method we have provided comparisons against other methods with a post hoc analysis. Results of mean scores are included in the appendix. We thank the reviewers and chair for encouraging us to explore this problem.

---

### Meta-Review · Area_Chair_ThYe · 2023-04-10

**Recommendation:** Invite to revise
**Confidence:** 3

**Metareview:**

The paper proposes a new perspective for drift detection. The idea of using diversity within an ensemble to detect drift seems intriguing, something that both I and reviewer d6SQ can appreciate.

However, I agree with Reviewer LAyt that anonymizing the main work that the paper compares its methodology to undermines our ability to review the paper — how can reviewers verify the claims of the author if we cannot read the paper that provides the contrast to this approach? A principle of science is to never take something on faith if possible. I can understand the motivation to anonymize; and can commiserate that it is sometimes unclear when submitting a blinded paper what to do.

For peer review, I'm not sure one should ever anonymize references (at all, I imagine, although perhaps there are edge cases I can’t think of), especially those that are necessary for reviewers to verify or understand one’s argument. Instead, you can write the paper such that authorship of particular papers is not claimed in a first-person way (e.g. you can say this paper builds off of X, instead of this paper builds off of our previous work X). For example, when reading this paper, that you were the authors of the paper you are contrasting to does not seem essential to the paper’s message. It is true that reviewers may sometimes guess that someone building off a framework may indeed be the author of that framework, but that is generally acceptable (I suppose it is just an aspect of scientific review culture) as long as one is attempting as best as one can to honor the spirit of blinding a paper.

Also, it appears as if the paper provides a method but does not have any experimental results? At least I could not find any within the PDF I was able to download. Both reviewers seem to be a bit confused by this (e.g. “what model and data are the authors using” and “It is assumed the proposed C2D2 framework is evaluated on synthetic datasets”). Perhaps there is a reason to propose the method without testing it, but that should likely be explicitly acknowledged and the reason for not running the algorithm should be explained. It is hard to review a proposed method without any results to contextualize what it does in practice — you could perhaps design a simple artificial data set at least as a proof of concept?

So unfortunately, I think that this paper cannot really be CCR in its current state (due centrally to the anonymization of the previous paper, and also due to the lack of empirical results), but the ideas seem interesting and I encourage the authors to keep developing them. And I feel bad that the anonymization approach is one of the main reasons for my recommendation, I can imagine it being disappointing -- my recommendation would be to, if possible, seek a few trusted people (perhaps a few elder grad students) to ask what to do when given a tricky situation with anonymization.

**Summary:**

The paper proposes a new method for drift detection that leverages an ensemble of diverse models trained on different subsets of the input and output space. The concepts seem interesting, but anonymizing the method compared to undermines peer review; also the lack of empirical results is not explained.

**Comments And Feedback To The Authors:**

One additional piece of feedback, is that a minor confusion I had when reading the paper (which could be a result of my own ignorance) is that when I looked up FCBF through the chain of references, it seemed like a method to reduce a feature set rather than to provide a score (that would go in the matrix that you would apply SVD to) — perhaps there is a way to clarify what metric within FCBF is being applied?

**Reason For Not Giving A Higher Recommendation:**

The main reason for aligning with the lower review score is that it isn't possible to evaluate some of the claims of the paper without knowing the details of the method being compared to. Also, while it can be okay to have a paper that proposes a method without any empirical results, usually it requires a strong argument -- otherwise the correctness, clarity, and reproducibility can become somewhat questionable. Here, it seems as if the method is almost fully specified, and at least perhaps an artificial data set could be constructed to grant reviewers confidence that the method as proposed does what it is designed to do.

**Reason For Not Giving A Lower Recommendation:**

N/A

---

### Decision · Program_Chairs · 2023-04-10

Revision accepted; invite to archive

---

> ### Author Response · Authors · 2023-06-16
> **Opt-in for archival**
>
> The authors agree to opt-in for archival.